# Understanding the Interplay between COX-2 and hTERT in Colorectal Cancer Using a Multi-Omics Analysis

**DOI:** 10.3390/cancers11101536

**Published:** 2019-10-11

**Authors:** Georgios D. Ayiomamitis, George Notas, Thivi Vasilakaki, Aikaterini Tsavari, Styliani Vederaki, Theodosis Theodosopoulos, Elias Kouroumalis, Apostolos Zaravinos

**Affiliations:** 1Laboratory of Gastroenterology Research, University of Crete, School of Medicine, 71013 Heraklion, Greece; g.agiomamitis@tzaneio.gr (G.D.A.); gnotas@med.uoc.gr (G.N.); kouroum@med.uoc.gr (E.K.); 21st Department of Surgery, Tzaneio General Hospital, 18536 Piraeus, Greece; s.vederaki@tzaneio.gr; 3Laboratory of Experimental Endocrinology, University of Crete, School of Medicine, 71013 Heraklion, Greece; 4Department of Pathology, Tzaneio General Hospital, 18536 Piraeus, Greece; th.vasilakaki@tzaneio.gr (T.V.); a.tsavari@tzaneio.gr (A.T.); 52nd Department of Surgery, Aretaieion Hospital, Medical School, National and Kapodistrian University of Athens, 11528 Athens, Greece; ttheodosop@med.uoa.gr; 6Department of Gastroenterology and Hepatology, University Hospital of Heraklion, 71013 Heraklion, Greece; 7Department of Life Sciences European University Cyprus, Nicosia 1516, Cyprus

**Keywords:** colorectal cancer, COX-2, PGE_2_, hTERT

## Abstract

Background: Cyclooxygenase 2 (COX-2) is involved in the initial steps of colorectal cancer (CRC) formation, playing a key role in the catalysis of arachidonic acid to prostaglandin E2 (PGE_2_). The human telomerase reverse transcriptase (hTERT or TERT) also plays an important role in colorectal cancer growth, conferring sustained cell proliferation and survival. Although hTERT induces COX-2 expression in gastric and cervical cancer, their interaction has not been investigated in the context of CRC. Methods: COX-2, PGE_2_ levels, and telomerase activity were evaluated by immunohistochemistry, ELISA, and TRAP assay in 49 colorectal cancer samples. PTGS1, PTGS2, PTGES3, TERT mRNA, and protein levels were investigated using RNA-seq and antibody-based protein profiling data from the TCGA and HPA projects. A multi-omics comparison was performed between PTGS2 and TERT, using RNAseq, DNA methylation, copy number variations (CNVs), single nucleotide polymorphisms (SNPs), and insertions/deletions (Indels) data. Results: COX-2 expression was positive in 40/49 CRCs, bearing cytoplasmic and heterogeneous staining, from moderate to high intensity. COX-2 staining was mainly detected in the stroma of the tumor cells and the adjacent normal tissues. PGE_2_ expression was lower in CRC compared to the adjacent normal tissue, and inversely correlated to telomerase activity in right colon cancers. COX-1 and COX-2 were anticorrelated with TERT. Isoform structural analysis revealed the most prevalent transcripts driving the differential expression of PTGS1, PTGS2, PTGES3, and TERT in CRC. COX-2 expression was significantly higher among B-Raf proto-oncogene, serine/threonine kinase, mutant (BRAF^mut^) tumors. Kirsten ras oncogene (KRAS) mutations did not affect COX-2 or TERT expression. The promoter regions of COX-2 and TERT were reversely methylated. Conclusions: Our data support that COX-2 is involved in the early stages of colorectal cancer development, initially affecting the tumor’s stromal microenvironment, and, subsequently, the epithelial cells. They also highlight an inverse correlation between COX-2 expression and telomerase activity in CRC, as well as differentially methylated patterns within the promoter regions of COX-2 and TERT.

## 1. Introduction

Colorectal cancer (CRC) is the third most common cancer worldwide among both sexes [1]. Although its incidence and mortality rates have been declining for several decades, a deeper understanding of the cellular mechanisms involved in it is imperative. Epidemiological studies have shown a 40–50% reduction in the risk of colorectal cancer in people who were chronically and systematically receiving nonsteroidal anti-inflammatory drugs (NSAIDs) [2,3,4], indicating their chemoprotective or chemotherapeutic influence. NSAIDs were previously shown to reduce the number and size of colorectal adenomas in patients with familial adenomatous polyposis (FAP) [5]. Most of the NSAIDs that are used today inhibit the action of prostaglandin endoperoxide H synthases 1 and 2 (PGHS-1 and PGHS-2; also known as PTGS-1 and -2 or cyclooxygenases-1 and -2, COX-1/-2, respectively). These are intracellular enzymes that catalyze the conversion of arachidonic acid into prostaglandins (PGs) and other eicosanoids. Both isoforms are involved in pain, fever, inflammation, and tumorigenesis [6] and constitute well-known NSAID drug targets.

COX-1 is expressed in almost every tissue with administrative-like effect, while COX-2 is a rapidly induced enzyme, regulated by specific stimulatory events. Under normal circumstances, COX-2 expression is limited to specific organs such as the central nervous system [7], the kidneys [8], and the eyes [9]. Following the replication process, COX-2 expression can dramatically increase, activating proinflammatory cytokines, growth factors, or tumor initiators [10]. COX-2 is induced by cytokines, growth factors, oncogenes, and tumor promoters, and has been found in high levels in breast [11], gastric [12], lung [13], prostate [14], urinary bladder [15], esophagus, pancreas, skin, oral, and colorectal cancers [16,17,18]. COX-2 overexpression alters cell adhesion, inhibits apoptosis, and modifies the response to growth regulatory signals [19]. It also prolongs the survival of abnormal cells, thereby favoring the accumulation of sequential genetic changes in the tumor [19]. Recent studies show that tumor stromal cells contribute to COX-2 upregulation in CRC. Increased levels of COX-2 were also detected in macrophages [20], stromal fibroblasts [21], and vascular endothelial cells [22], indicating that the host and tumor cells may contribute to the production of prostaglandin within the tumor microenvironment and subsequently, to the development of cancer.

Telomerase is an RNA-dependent DNA polymerase that adds telomeric DNA at the chromosome termini. The core telomerase complex consists of two components: the catalytic subunit telomerase reverse transcriptase (TERT or hTERT in humans) and the ubiquitously expressed telomerase RNA component (hTERC) [23]. Most normal human cells lack telomerase activity because of the stringent transcriptional repression of the hTERT gene, whereas the induction of hTERT expression and telomerase activation is, in general, a prerequisite step for the malignant transformation of human cells [23]. Knocking-down hTERT upregulates the expression of COX-2 in hTERT-expressing gastric and cervical cancer cells [24]. On the other hand, hTERT knockdown inhibits the growth of pancreatic cancer cells via downregulation of COX-2 [25]. Nevertheless, the interaction between hTERT and COX-2 has not been investigated in the context of colorectal cancer.

In this study, we biopsied tissues from patients diagnosed with colorectal cancer, and studied the differences in COX-2 expression between epithelial and stromal cells of tumor and adjacent normal tissues. We measured the expression of prostaglandin E_2_ (PGE_2_) in these tissues, compared it with hTERT activity, and associated it with various clinicopathological parameters of the CRC patients. We further investigated the expression of COX-1, COX-2, PTGES3, and TERT genes, using bioinformatics analysis on the TCGA-COAD and READ datasets, and found that COX-1 and COX-2 are mainly correlated to TERT expression, and that high COX-2 expression is associated with a better overall survival of the CRC patients. Importantly, we provide evidence that the promoter regions of the PTGS2 and TERT genes are reversely methylated, but this does not seem to affect the reverse expression levels of these genes.

## 2. Results

### 2.1. Immunohistochemical Expression of COX-2

COX-2 protein expression was positive in the epithelial cells of 40/49 (81.6%) colorectal cancer tissue specimens. Of these, 22/49 (44.8%) samples had high COX-2 staining (>50% of epithelial cancer cells), whereas 14/49 (28.6%) samples had medium (≤50% and >25% of epithelial cancer cells) and 5/49 (10.2%) samples, low (≤25% of epithelial cancer cells) COX-2 staining. On the other hand, 9/49 (18.4%) CRC samples were negative for COX-2. The dye reaction was cytoplasmic, heterogeneous and its intensity was moderate to strong (Figure 1A–D and Table 1). Staining was also detected in the stromal cells of the tumor (containing mast cell, fibroblasts, and endothelial cells), but in a significantly lower percentage compared to the cancerous epithelial cells (Figure 1E). Of these, 20/49 (40.8%) cancer samples stained low (≤25% of stromal cancer cells), 15/49 (30.6%) stained at a lower percentage (≤15% of stromal cancer cells), and 5/49 (10.2%) even less (≤5% of stromal cancer cells) at the tumor’s stroma. Importantly, COX-2 expression in the tumor’s stromal cells increased proportionally with the corresponding expression in the tumor’s epithelial cells. As a result, in cancer tissues where COX-2 epithelial expression was >50%, higher COX-2 levels were observed in the tumor’s stroma by enhanced staining of mast cells, fibroblasts, and endothelial cells (Figure 1F). We also observed increased COX-2 protein expression in the stroma of the adjacent normal tissues, which was proportional to the corresponding percentage of COX-2 staining in the stroma of cancerous tissues (10–15%) (Figure 1G). COX-2 protein expression was observed at low levels (~3%) in the epithelium of the adjacent normal tissues as well (Figure 1H).

### 2.2. PGE_2_ Composition and PTGS1, PTGS2, PTGES3, TERT Expression in CRC

PGE_2_ expression was evaluated in the colorectal cancer samples, along with their corresponding adjacent normal tissue. PGE_2_ levels were significantly lower in the cancer samples compared to their corresponding adjacent normal tissue (1550 ± 500 pg/mL vs. 2500 ± 500 pg/mL, *p* < 0.013) (Figure 2A). This was also supported by the significantly lower PTGS1 and PTGS2 levels in the COAD and READ tumors, when compared to the normal tissue samples (from TCGA and GTEx). On the other hand, the mRNA levels of PTGES3, which converts prostaglandin endoperoxide H_2_ (PGH_2_) to prostaglandin E_2_ (PGE_2_), were significantly higher in the COAD and READ tumors as opposed to their normal counterparts. Likewise, hTERT exhibited higher expression in the COAD and READ tumors compared to the normal tissue (Figure 2B). This reverse association between PGE_2_ levels and hTERT activity was further noted in our patient cohort. Specifically, hTERT activity dropped from the right to the left colon, and the rectum cancers; conversely, PGE_2_ levels increased respectively (*p* = 0.006) (Figure 2C). Furthermore, hTERT expression was inversely correlated with PTGS1 (*p* = 1.7 × 10^−16^, *R* = −0.32) and PTGS2 (*p* = 0.0086, *R* = −0.11) expression, but not with PTGES3 expression (*p* = 2 × 10^−5^, *R* = 0.17) in the TCGA-COAD dataset (Figure 2D). Taken together, these data highlight the reverse association between COX-1/-2 expression and hTERT activation in colorectal cancer.

In addition, we explored the protein levels of COX1, COX-2, and TERT using the tissue microarrays (TMA) from the Human Protein Atlas (HPA). Regarding COX-1, we found medium staining (with cytoplasmic/membranous location) in 1/10 (10%) CRC samples, low staining in 5/10 (50%) samples, and no staining in 4/10 (40%) samples. The intensity of COX-1 was moderate in 2/10 (20%) samples, weak in 7/10 (70%) samples, and negative in 1/10 (10%) CRC sample. The respective quantity levels were >75% in 5/10 (50%) samples, <25% in 4/10 (40%) samples, and none in 1/10 (10%) samples (Figure 3A–C). 

The respective COX-2 staining in the TMA was medium (with cytoplasmic/membranous location) in 1/12 (8.3%) sample, low in 4/12 samples (33%), and absent in 7/12 CRC samples (58.3%). The intensity was strong in 1/12 (8.33%) sample, moderate in 2/12 (16.6%) samples, weak in 4/12 (33.3%) samples, and negative in 6/12 (50%) samples. The respective quantity levels ranged from <25% in 3/12 samples, <25% in 3/12 samples, and none in 6/12 samples (Figure 3D–F). On the other hand, contrary to the RNAseq results, TERT protein expression was absent in all CRC samples in the TMA (11/11, 100% negative) (Figure 3G–I).

In addition, the PGE_2_ levels did not differ between CRC samples in terms of sex, age, smoking, body mass index (BMI), or tumor topography. Interestingly, we found significantly higher PGE_2_ levels in the adjacent normal tissue of patients with Dukes stage A or B cancer, compared to those in Dukes stage C or D (*p* = 0.036, *t*-test). In addition, we found no difference in the levels of PTGS1 or PTGS2 among tumors of different staging (*p* > 0.05), but we found lower PTGES3 levels and higher TERT levels in tumors of higher stage (III and IV), compared to lower stage tumors (I and II) (Table 2 and Appendix A). These data indicate that during the initial stages of colorectal cancer development, the normal microenvironment of the patients is more susceptible to tumor build-up through the COX-2 pathway, while at an advanced stage of the disease, the cancer spread may reduce this ability. Also, the regenerative ability of the epithelium in the adjacent normal tissue of patients with advanced colorectal cancer may be lower. Finally, in advanced disease, decreased COX-2 expression is observed, along with arachidonic acid deprivation, due to the catalysis mediated by COX-2. Taken together, these explain the low PGE_2_ levels in the CRC tumors.

COX-2 overexpression is associated with an unfavorable overall survival of patients with urinary bladder cancer [26]. However, contrasting results exist regarding whether COX-2 can predict the outcome of CRC patients after surgery [27]. On the other hand, it is unknown whether the expression of PTGS1 and PTGES3 is associated with patient survival. To this end, we examined whether PTGS1 (COX-1), PTGS2 (COX-2), and PTGES3 can be used as independent predictors in the CRC patients’ outcome, using the TCGA-COAD and READ datasets. We found that the expression of PTGS1 and PTGES3 is unrelated to CRC patient survival. On the other hand, high PTGS2 mRNA expression is significantly associated with a better overall (but not disease-free) survival among CRC patients (*p* = 0.0078, log-rank test) (Figure 4A–D).

Principle component analysis (PCA) was used to reduce the dimensionality of the TCGA-COAD and READ data sets for the tumor and normal samples, as well as for sigmoid and transverse normal colon samples, based on the expression of PTGS1, PTGS2, PTGES3, and TERT. Using this analysis, colorectal tumors (COAD and READ) clustered separately from the corresponding normal tissues, as well as from the normal sigmoid and transverse colon tissue from the GTEx project. (Figure 4E,F).

### 2.3. Isoform Gene Expression Analysis

COX-2 expression was previously reported to be markedly elevated in CRC compared to the normal mucosa [28]. Although this conflicts with our results, which indicated both PTGS1 and PTGS2 downregulation in CRC, we examined whether such a discrepancy could be attributed to differential expression among the gene isoforms. To this end, we performed isoform structural analysis of PTGS1 (ENSG00000095303.14), which showed most prevalent expression of the PTGS1−001, PTGS1−002, and PTGS1−203 transcripts. As the PTGS1−003 transcript lacks an “EGF” and an “An_peroxidase” domain, it exhibited the lowest expression levels in both COAD and READ tumors. Likewise, the most prevalent PTGS2 transcript (ENSG00000073756.11) was PTGS2−001, followed by PTGS2−003 and PTGS2−002. Also, the most abundant PTGES3 (ENSG00000110958.15) transcripts were PTGES3−001, PTGES3−003, and PTGES3−004, as PTGES3-002 lacks a “CS domain” (CHORD-containing proteins and SGT1 [29], a ~100-residue protein–protein interaction module). Regarding TERT (ENSG00000164362.18), the most highly expressed isoforms were TERT−001, TERT−003, and TERT−007. The TERT-005 and TERT-006 isoforms contain a “telomerase RNA binding domain (RBD)”, but lack the “RVT_1” domain. (Figure 5). These data suggest that differential expression of specific isoforms is related to the expression of COX-1, COX-2, COX-3, and TERT in colorectal cancer.

### 2.4. Multi-Omics Analysis

We further evaluated whether PTGS2 and TERT expression differs between KRAS or BRAF mutant and wild type CRC tumors. Examining the RNA-seq and mutational data among 396 tumors within the TCGA-COAD dataset, we found that KRAS mutations did not seem to affect the expression of neither PTGS2 nor TERT. Nevertheless, PTGS2 levels were significantly higher among BRAF^mut^ COAD tumors (carrying mainly missense/in frame and a few deleterious mutations), compared to the BRAF^wt^ ones (*p* < 0.0001) (Figure 6). 

Post-translational modifications were recently reported to regulate COX-2 activity, as well as its intracellular localization and stability in CRC [30]. We stepped in to further investigate the interplay between the differential gene expression, promoter methylation, copy number aberrations, and somatic mutational (SNPs and small INDELs) profiles in the COX-2 and TERT genes in the TCGA-COAD dataset. Interestingly, we found a broad hypomethylation within the promoter region of PTGS2 and an extensive hypermethylation in the corresponding region of TERT (Figure 7 and Appendix A). Nevertheless, the methylation profiles did not seem to differ between high and low PTGS2- (or TERT-) expressing colorectal cancer samples, suggesting that the reverse expression profiles between PTGS2 and TERT are probably not due to the reverse methylation profiles found within their promoter regions.

Taken together, our findings provide a comprehensive understanding of the interaction between the cyclooxygenase genes and hTERT in CRC and highlight the inverse correlation between COX-2 expression and the activity of telomerase. These outcomes may help guide patient selection and identify the optimal combination of therapeutic regimens to enhance therapies and clinical response in colorectal cancer patients.

## 3. Discussion

Growing interest has now focused on the ability of COX-2 to suppress apoptosis, as this is thought to favor carcinogenesis by allowing the survival of mutant cells. There are several hypotheses explaining the reason for the suppression of apoptosis in response to COX-2 upregulation. These include prostaglandin-induced overexpression of bcl-2 [31], and decreased arachidonic acid, which is an inducer of apoptosis, through its COX-2-induced transformation into prostaglandin [32]. Conversely, many NSAIDs increase apoptosis. Nevertheless, this is unlikely to occur only via suppression of COX-2 activity, since NSAID-induced apoptosis has been observed in cells that do not express cyclooxygenase [33]. Additionally, studies have shown that several factors that stimulate COX-2 expression also induce apoptosis.

COX-2 appears to play a major role in carcinogenesis in various systems, such as in the colon, breast, lung, and pancreas [13,16,34,35,36,37]. COX-2 inhibitors, including aspirin, celecoxib, and other NSAIDs, seem to be effective in colorectal cancer prevention [38], suggesting a pathogenic role of COX-2 in the development of CRC [27]. PGE_2_ along with other prostaglandins, seem to play a significant role in the tumor onset, affecting mitogenesis, cell adhesion and penetration, and apoptosis [39]. However, unlike COX-2, PGE_2_ has not been widely studied, probably because it is an unstable product with a half-life of 30 seconds, rendering it difficult to be isolated and studied.

Previous reports showed that COX-2 protein is selectively expressed in cancer cells, but not in the tumor stroma [34,40]. In contrast, other studies stressed the requirement of COX-2 to be expressed both in the stroma and the epithelium of cancer cells, adenomas [41,42], and carcinomas [43,44,45]. In our study, a positive COX-2 immunoreaction was detected by cytoplasmic staining in the majority of the epithelial cancer cells. We also found, to a lesser extent, a positive immunoreaction against COX-2 in the stromal cancer cells, including mononuclear cells, fibroblasts, endothelial cells, and smooth muscle cells, as previously reported [46]. This demonstrates the ability of both cell types to contribute to the tumor formation. Furthermore, it is worth noting that although we detected intense COX-2 immunostaining in the majority of the epithelial cancer cells, there was an inconsistency in the COX-2 mRNA expression derived from the analysis of the TCGA-COAD dataset. This is probably because the TCGA data do not differentiate between epithelial and stromal cells. A similar disproportion, probably attributed to the different antibodies used and/or different sample numbers, was also noted between our COX-2 staining results and those derived from the HPA’s TMA slides, the majority of which (7/12 CRCs, 58.3%) did not stain for COX-2 and none for TERT. Such a discrepancy could also be plausible because IHC on the TMA slides does not differentiate between epithelial and stromal cells and does not take into account the staging of the disease (Appendix A). 

Charalambous et al. [47] found significantly increased COX-2 levels in epithelial cells of the tumor, compared to adjacent normal epithelium, but not in the stromal cells. COX-2 expression was also found to be higher in stromal cells of the adjacent normal tissue compared to the stroma of the tumor [48]. We also found similar results, as well as lower PGE_2_ composition in the cancerous tissue compared to the adjacent normal mucosa. In addition, we found lower COX-1 and COX-2 expression in the colorectal cancer compared to the adjacent normal tissue. These results prove that within the tumor’s microenvironment, COX-2 expression is mainly epithelial, while, in the adjacent normal tissue, it is stromal. Therefore, in line with other studies, we suggest that COX-2 plays a major role during the initial stages of colorectal carcinogenesis, and that in stromal cells, COX-2 expression is directly implicated in angiogenesis, by preparing the adjacent normal tissue for the local growth and progression of the malignant tumor [49,50].

COX-2 expression is mediated by various factors, such as NSAIDs [51], TGF [52], butyrate [53], and ceramide, suggesting that its upregulation might be attributed to apoptosis. COX-2 stimulation is also inversely proportional to the nuclear site of the catalytic component of hTERT [26]. Nevertheless, the regulatory factors driving COX-2 expression in the local setting remain unknown.

Here we provide evidence that PGE_2_ levels are significantly lower in the tumor compared to the adjacent normal tissue, advocating that COX-2 has a key role in the early stages of CRC development. We detected significantly higher PGE_2_ levels in left compared to right colon tumors, indicating the different behavior according to the tumor’s topography. In addition, we show that PGE_2_ levels are reversely correlated with hTERT activation. In the right colon tumors, we found decreased PGE_2_ levels and increased hTERT expression, which is indicative of suppressed apoptosis. In contrast, in left colon and rectum cancers, we found higher PGE_2_ levels accompanied by decreased hTERT activation, which is indicative of enhanced apoptosis. In support of our observation, studies have previously reported COX-2 upregulation in rectal, but not colon cancer [54]. 

We also show that high COX-2 expression is associated with a better overall survival among CRC patients. Our analysis reveals that the differential expression of COX-1, COX-2, COX-3, and TERT is mainly due to specific transcripts in the disease. Interestingly, our data reveal that COX-2 expression is significantly higher among BRAF^mut^ colon adenocarcinomas, compared to BRAF^wt^ tumors. We finally provide evidence that the promoter regions of the genes PTGS2 and TERT are reversely methylated; however, this does not seem to affect the expression levels of these genes in CRC.

## 4. Materials and Methods

### 4.1. Patients

Surgical specimens of primary colon and rectal adenocarcinomas with informed consent were obtained from 49 patients (28 males and 21 females aged 49–87 years, mean age 64.0 years ± 1.63 SEM) operated at the Tzaneio General Hospital, Piraeus, Greece, between 2002 and 2004 (Table 3). All specimens were histologically verified for colorectal cancer. Informed consent was obtained from all patients and ethical approval for the study was obtained from the Human Research Ethics Committee of the Tzaneio General Hospital (TGH#16527/4-12-2017). Immediately after removal of the surgical sample, four small tissue blocks were collected from each cancer sample and four others from the seemingly healthy adjacent colon mucosa, at ~10 cm from the tumor’s center. They were then immediately immersed into liquid nitrogen and kept at −80 °C, and the remaining sample was sent for pathological assessment.

### 4.2. Prostaglandin E_2_ Expression

The expression of Prostaglandin E_2_ (PGE_2_) was evaluated using the Prostaglandin E_2_ ELISA kit - Monoclonal (Cayman Chemical Co., Ann Arbor, MI, USA), following the manufacturer’s instructions. Snap-frozen tissue samples (100 mg) were homogenized on a sonicator and divided into two parts. After being left for 30 min at 4 °C, 100 μL of the first part were centrifuged at 12,000 *g* for 5 min at 4 °C and the supernatant was rapidly stored at −80 °C. We used part of this material in order to calculate the total protein following the Bradford (Bio-Rad Laboratories, Basel, Switzerland) protocol and using bovine serum albumin (BSA) as an invariant. The second part of the homogenate was used for the isolation of PGE_2_. To precipitate proteins, 1 mL of ethanol was added to 250 μL of the second part, vortexed, incubated at 4 °C for 5 min, and centrifuged at 3000× g for 10 min at 4 °C to remove precipitated proteins. The supernatant was collected, ethanol was evaporated by centrifugation, and acetate buffer (pH = 4.0) was added to acidify the sample to a pH ~ 4.0. Next, 5 mL of methanol 1% and ethyl acetate were added to each sample. and PGE_2_ was extracted by passing through SPE Cartridge (C18) columns, using gravity. Ethyl acetate and methanol were finally vaporized in a vacuum centrifuge. The ELISA buffer was added to resuspend the samples, achieving 1:10 and 1:100 thinning (depending on the protein concentration in each sample that was measured earlier by Bradford). PGE_2_ concentration was measured by ELISA at 405–410 nm. All samples were measured in triplicates and the analysis was repeated three times. A positive control was used to create a standard curve consisting of eight serial dilutions, ranging from 1000–10 pg/mL PGE_2_. A “cold spike” was used to calculate the sample’s purity (%).

### 4.3. Immunohistochemistry (IHC)

For each patient sample, 4 mm thick incisions of paraffin blocks containing cancerous tissue were used for immunohistochemical examination of COX-2 expression. Staining was conducted with immunoperoxidase in three stages using the Dako kit. Initially, the samples were deparaffinized with xylose and rehydrated in different concentrations of alcohol. Endogenous peroxidase was inhibited using hydrogen peroxide (H_2_O_2_) 0.3% in Tris buffer (pH 7.6) for 15 min. Before the effect of the primary antibody, the incisions were immersed in 10nM of citrate buffer, rinsed in Tris-buffered saline (TBS), and warmed up in a microwave oven (650–800 W) for 30 min. To avoid nonspecific links, the incisions were rinsed with TBS before adding the primary antibody (mouse monoclonal anti-COX-2, 1:100 dilution, Novocastra, Newcastle upon Tyne, UK). Tumors with previously known high levels of COX-2 expression were used as positive controls. Presensitized adsorbed serum was used as a negative control to verify the non-specific staining. Two independent pathologists, who were not given the clinical information, evaluated the immuno-reaction, specifying the percentage of positive colored cancer cells. Differences were resolved with consensus. The staining was confined to the cytoplasm and perinuclear area. Positively stained cells were counted in five randomly selected fields at magnification of ×400. The total number of positive cells/100 cells per case was calculated. All cases were classified in three categories according to the intensity of COX-2 staining, as mild (category I, ≤25%), moderate (category II, >25% and ≤50%), and intense (category III, >50%). The percentage in each category corresponds to the percentage of positively colored neoplastic cells. The intensity of staining was assessed as weak, medium, or strong. The choice of the lowest limit for each separate marker was based on the corresponding categorization found in other studies. The choice of threshold was previously found to characterize better predictive clinicopathological and molecular correlations of the indicators under study.

### 4.4. Telomeric Repeat Amplification Protocol (TRAP)

The TRAP in vitro assay (Quantitative Telomerase Detection Kit-US Biomax, Inc., Rockville, MD, USA) was used to detect telomerase activity in the cancer patient samples, as previously described [55]. In detail, samples were lysed and the telomerase activity was determined through its ability to synthesize telomeric repeats onto an oligonucleotide substrate in vitro. The resultant extended products were amplified by qPCR using SYBR Green, on a CFX96 Touch real-time PCR thermal cycler (Biorad, CA, USA). A standard curve using a positive control (TSR template) with a range from 0.5 μg/μL (3 × 10^5^ molecules/reaction) to 6.4 × 10^−6^ μg/μL (4 molecules/reaction) was used to estimate telomerase activity.

### 4.5. TCGA Data Extraction and RNA-Seq Analysis

RNA-seq data (read counts) were extracted from the Cancer Genome Atlas (TCGA) COAD and READ datasets (275 colorectal cancer samples) using the Genomic Data Commons (GDC) Data Portal (https://portal.gdc.cancer.gov/) and normalized to transcripts per million (TPM) mapped reads [56,57]. Briefly, read counts were initially divided by the length of each gene in kilobases (reads per kilobase, RPK) and were then counted up and divided by 1,000,000 (“per million” scaling factor), producing the TPM values for each gene, in each tumor sample. A small offset was added to avoid taking log of zero.

Specifically, the mRNA expression levels of the genes prostaglandin-endoperoxide synthase 1 (PTGS1 or COX-1), prostaglandin-endoperoxide synthase 2 (PTGS2 or COX-2), prostaglandin E synthase 3 (PTGES3), and telomerase reverse transcriptase (TERT) were evaluated using the “limma” R package with the following cutoffs: log_2_FC = 1 and *q*-value = 0.01. Gene expression was scaled in log_2_(TPM+1) values for all samples. TCGA normal samples were matched with normal samples from the Genotype-Tissue Expression project (GTEx) (http://www.gtexportal.org), totaling 349 normal COAD and 318 normal READ samples, respectively. The correlation between PTGS2 (COX-2) and TERT expression in both TCGA datasets, as well as in the GTEx Colon-Sigmoid and Colon-Transverse normal datasets, was evaluated using Pearson’s correlation (on the log-scale axis for visualization). Principal component analysis (PCA) was used to reduce the dimensionality of the TCGA-COAD, READ and GTEx data sets, based on the expression of PTGS1, PTGS2, PTGES3, and TERT in their samples, and to cluster them in a three-dimensional representation.

### 4.6. Isoform Analysis

The expression distribution of PTGS1, PTGS2, PTGES3, and TERT was profiled using the “easyGgplot2” R package. Each gene’s isoform usage was profiled using bar plots in each TCGA dataset (COAD and READ).

### 4.7. Survival Analysis

Overall and disease-free survival analysis (Kaplan–Meier plots) was performed using median cutoff and hazards ratio (HR), 95% confidence interval (CI) as a dotted line. Survival maps for the TCGA-COAD and TCGA-READ datasets were constructed with a significance level at *p* = 0.05 and FDR adjustment (log-rank test).

### 4.8. Data Extraction from the Human Protein Atlas (HPA) and Protein Expression Analysis

Protein expression data were retrieved from antibody-based protein profiling using IHC from the Tissue Atlas of the Human Protein Atlas (HPA). Information regarding the protein distribution of PTGS1, PTGS2, and TERT across all colorectal cancer tissues was retrieved from HPA’s Tissue [58]. The antibodies used for IHC in the respective TMAs, were as follows: HPA002834, anti-PTGS1; HPA001335, anti-PTGS2; HPA054641, anti-TERT.

### 4.9. Statistical Analysis

The statistical analysis was performed using the IBM SPSS statistics software, version 19 (ibm.com/support/pages/spss-statistics-190-fix-pack-1). The results were expressed as mean ± standard errors. Depending on applicability, the *t*-test (paired and unpaired) and One-way ANOVA test were used, with Bonferroni post hoc analysis of differences between groups. The χ^2^ test was used for the analysis of non-parametric data in 2 × 2 tables. The Pearson and Spearman tests were used for correlation analyses using a threshold of *p* = 0.05.

### 4.10. Multi-Omics Analysis

The UCSC Xena tool was used to explore the multi-omics and clinical/phenotypic data of the GDC-TCGA-COAD and READ datasets [59]. Gene expression (RNAseq) was evaluated using the normalized HTSeq-Fragments Per Kilobase of transcript per Million mapped reads upper quartile (FPKM-UQ), or log2 (TPM + 0.001) to compare gene expression levels between the GTEx and TCGA studies; DNA methylation was analyzed with Illumina’s Human Methylation 450; copy number aberrations with masked copy number segment; and somatic mutations (deleterious, splice, missense/in frame, silent and complex or unannotated mutations) were estimated using MuTect2 variant aggregation and masking.

## 5. Conclusions

Taken together, the data from our study provide a comprehensive understanding of the interactions between cyclooxygenases and hTERT in colorectal cancer and may help guide patient selection and identify the optimal combination of therapeutic regimens to enhance therapies and clinical response in cancer patients.

## Figures and Tables

**Figure 1 cancers-11-01536-f001:**
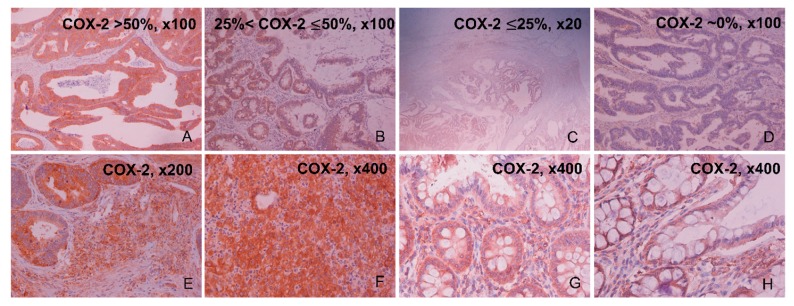
Representative slides of COX-2 staining in colorectal cancer (CRC) by immunohistochemistry (IHC). Colorectal adenocarcinomas with highly positive COX-2 immunoreaction (>50% of epithelial cancer cells; 22/49 (44.8%) CRC samples) (**A**), moderately positive (>25% and <50% of epithelial cancer cells; 14/49 (28.6%) CRC samples) (**B**), mildly positive (<25% of epithelial cancer cells; 5/49 (10.2%) CRC samples) (**C**), and negative COX-2 immunoreaction (9/49 (18.4%) CRC samples) (**D**). CRC slides were also imaged at the tumor’s stroma, containing mast cells, fibroblasts, pericytes, and endothelial cells. COX-2 positive stroma staining at ×200 (**E**) and ×400 (**F**). Adjacent to the tumor, normal slides were also imaged at the stroma region, with representative slides of COX-2 positive stroma staining at ×400 (**G**,**H**).

**Figure 2 cancers-11-01536-f002:**
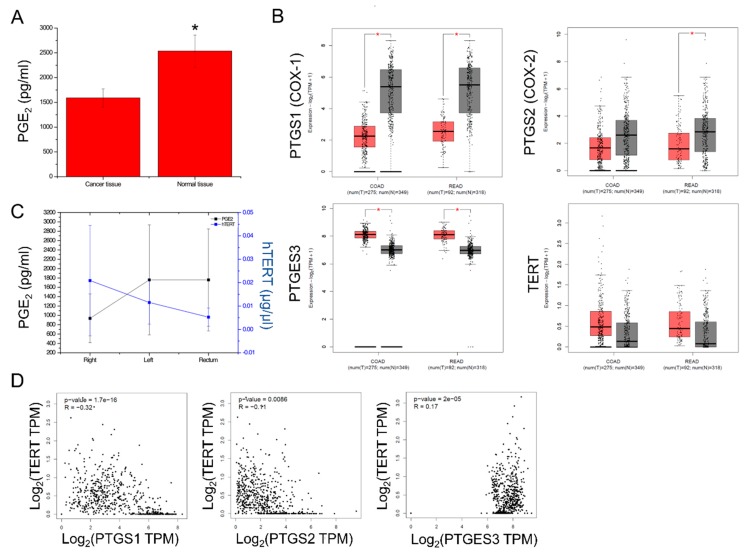
PGE_2_ levels were significantly lower in colorectal cancer (n = 49), compared to the adjacent normal tissue samples (* *p* < 0.01) (**A**). PTGS1 (COX-1) and PTGS2 (COX-2) mRNA levels were significantly lower across COAD and READ tumors compared to the normal mucosa (TCGA and GTEx). On the other hand, PTGES3 (COX-3) and TERT mRNA levels were significantly upregulated in the tumors (**B**). TERT activity decreased from right to left colon, and rectum cancers; conversely, PGE_2_ levels increased respectively (*p* = 0.006) (**C**). TERT expression was inversely correlated with the expression of PTGS1 and PTGS2, but not with the expression of PTGES3 in the TCGA-COAD dataset (**D**).

**Figure 3 cancers-11-01536-f003:**
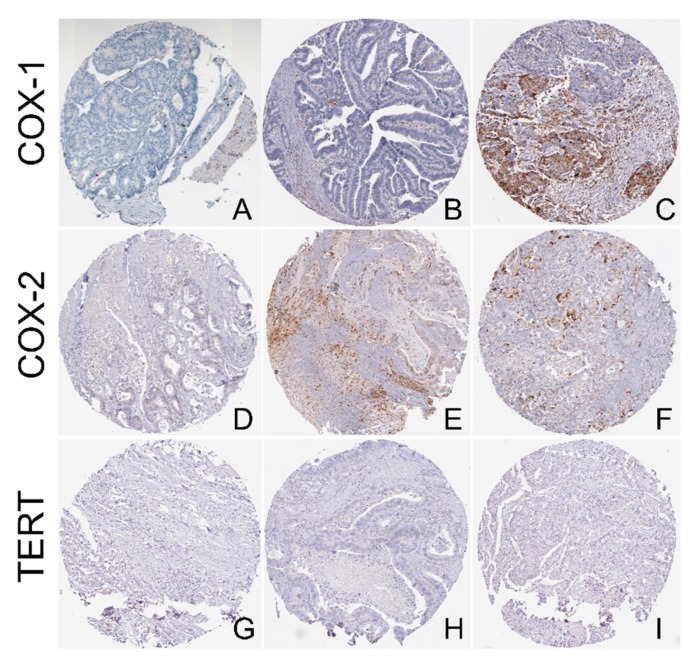
Representative tissue microarray (TMA) slides with COX-1, COX-2, and TERT-staining. (**A**) Negative COX-1 staining was detected in 4/10 (40%) CRC samples; (**B**) Low COX-1 staining was detected in 5/10 (50%) CRC samples; (**C**) Medium intensity COX-1 staining was detected in 1/10 (10%) CRC samples; (**D**) Negative COX-2 staining was detected in 7/12 CRC samples (58.3%); (**E**) Low COX-2 staining was detected in 4/12 CRC samples (33%); (**F**) Medium COX-2 staining was detected in 1/12 (8.3%) CRC samples; TERT staining was negative in all CRC samples (11/11, 100%) (**G**–**I**). All slides were retrieved from the Human Protein Atlas (HPA). Magnification, ×100.

**Figure 4 cancers-11-01536-f004:**
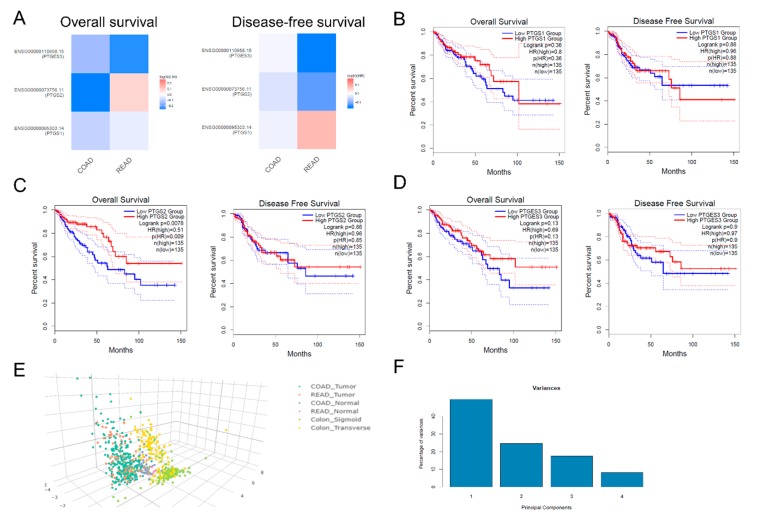
(**A**–**D**) PTGS1 and PTGES3 expression are is unrelated to CRC patient survival; however, PTGS2 expression is significantly associated with a better overall (but not disease-free) survival among CRC patients (*p* = 0.0078, log-rank test). (**E**,**F**) Colorectal tumors (COAD and READ) clustered separately from the corresponding normal tissues, as well as from the sigmoid and transverse normal colon tissue from the GTEx project, based on the expression of PTGS1, PTGS2, PTGES3, and TERT.

**Figure 5 cancers-11-01536-f005:**
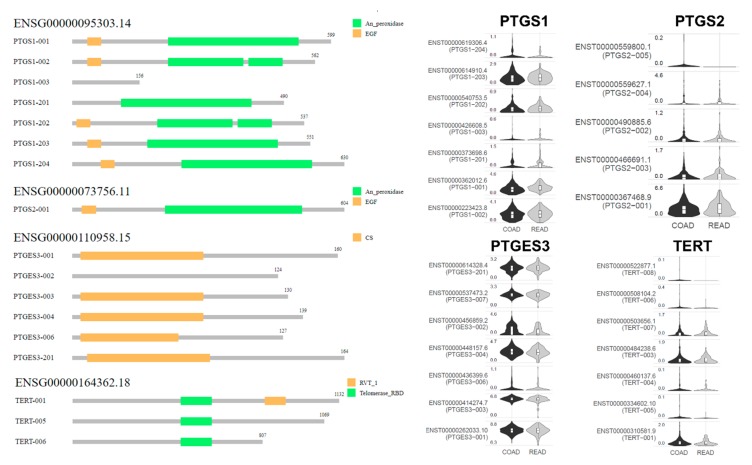
Isoform structural analysis shows that the differential expression of specific transcripts is related to the expression of the COX-1 (ENSG00000095303.14; PTGS1), COX-2 (ENSG00000073756.11; PTGS2), COX-3 (ENSG00000110958.15; PTGES3), and TERT (ENSG00000164362.18) genes in colorectal cancer.

**Figure 6 cancers-11-01536-f006:**
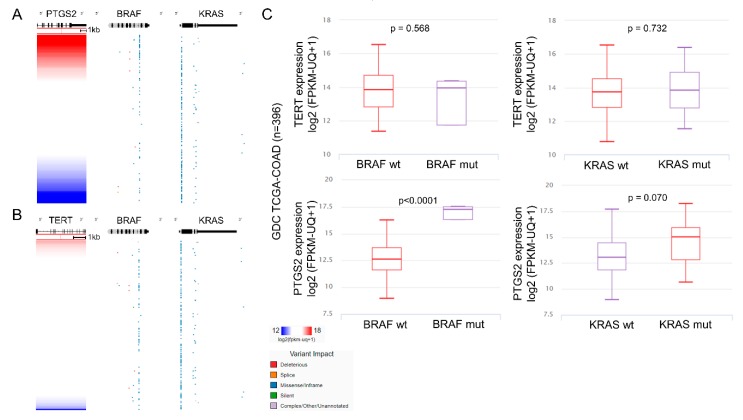
KRAS mutations do not affect the expression of PTGS2 (COX-2) (**A**) or TERT (**B**). PTGS2 (COX-2) expression, on the other hand, is significantly higher among BRAF^mut^ COAD tumors carrying mainly missense/in frame and a few deleterious mutations, compared to the BRAF^wt^ ones (**C**). KRAS, a Kirsten ras oncogene homolog from the mammalian ras gene family, encoding a protein belonging to the small GTPase superfamily; BRAF, B-Raf proto-oncogene, serine/threonine kinase; PTGS2, prostaglandin-endoperoxide synthase 2; TERT, telomerase reverse transcriptase.

**Figure 7 cancers-11-01536-f007:**
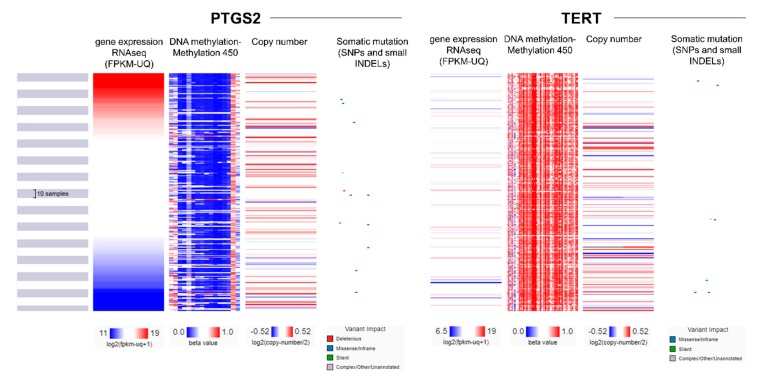
A multi-omics analysis revealed that the promoter of PTGS2 is broadly hypomethylated, whereas that of TERT is extensively hypermethylated in colorectal cancer. Nevertheless, no significant association was found between DNA methylation, copy number aberrations, or somatic mutations (SNPs and small INDELs) and the expression pattern of PTGS2 or TERT genes.

**Table 1 cancers-11-01536-t001:** COX-2 immunostaining in the CRC patient cohort (n = 49).

	Percentage (%) of Positive COX-2 Immunostaining by IHC			COX-2 Staining Intensity by IHC		
CRC Patient ID#	Low (≤25% of Epithelial Cancer Cells)	Medium (>25% & ≤50% of Epithelial Cancer Cells)	High (>50% of Epithelial Cancer Cells)	Weak	Moderate	Strong
1405E/03		+		+		
2571D/03	+				+	
1044G/03	+					+
2906E/04	+			+		
2908D/04			+			+
3661E/03			+			+
1404E/03			+			+
856E/03			+			+
2909D/04		+			+	
2831D/04		+				+
975D/02	Negative staining					
2904Z/04		+				+
948G/03			+			+
348E/03			+			+
5071D/03			+			+
988E/04		+				+
6003G/02			+			+
977E/03		+			+	
857E/03	Negative staining					
304D/03			+			+
3986E			+			+
1694G/03		+				+
1096		+		+	+	
1254G	Negative staining					
5433G	+		+		+	
633Z	+			+		
987E/04		+				+
423D/03		+			+	
656D/03			+			+
4961D/02			+			+
5025E/02			+			+
482E/03		+		+		
770E/03			+			+
3386E		+				+
1403D/03			+			+
5758G/02		+				+
5565G		+			+	
635D			+			+
3701D			+			+
3876D			+		+	
8122/03	Negative staining					
4975E/02	Negative staining					
5024Z			+		+	
5013D			+			+
5163Z/02	Negative staining					
5884E/02			+			+
5024Z/02	Negative staining					
5019D/03	Negative staining					
5199G/02	Negative staining					

**Table 2 cancers-11-01536-t002:** PGE_2_ levels (pg/mL) in colorectal cancerous tissue and its adjacent mucosa. Colorectal cancer patients have been stratified according to different clinicopathological characteristics. Statistically significant *p*-values are highlighted in bold.

Variable	Normal Mucosa				Cancer Tissue			
	No	Mean	SD	*p*-Value	No	Mean	SD	*p*-Value
**All cases**	35	2374.04	1695.081		34	1589.622	1075.305	0.013
**Sex**								
Male	13	1976.704	1477.7668		13	2034.405	1305.7355	
Female	22	2866.836	2109.3560	0.056	21	1314.280	822.6191	0.191
**Smoke**								
No	24	2374.842	1669.7945		24	1419.926	860.0917	
Yes	8	2796.588	2498.2338	0.59	7	1667.929	1206.1716	0.545
**BMI**								
<25	19	2244.542	1745.6862		19	1812.677	1191.0385	
≥25	16	2882.578	2124.8944	0.336	15	1307.085	865.2102	0.177
**Duke’s stage**								
A	0				0			
B	16	3273.561	2036.8244		15	1696.322	904.9637	
C	15	1930.935	1782.8859		15	1423.153	1294.7909	
D	4	1856.634	1055.7880		4	1813.756	900.6809	
A+B	16	3273.561	2036.8244	**0.036**	15	1696.322	904.9637	0.615
C+D	19	1915.293	1630.6619		19	1505.385	1210.7447	
**Grade**								
low	4	2617.679	1318.3896		4	1261.628	802.6919	
moderate	28	2682.486	2039.2589		27	1700.093	1127.0442	
high	3	1062.409	859.9787	0.393	3	1032.711	868.3637	0.495
**Tumor site**								
Right	7	1667.951	1410.0214		7	936.412	523.005	
Left	7	2929.321	2674.6937		7	2077.241	1396.8070	
Sigmoid	10	2580.901	1377.0804		10	1536.713	1007.5961	
Rectum	11	2797.966	2169.4357	0.236	10	1758.445	1090.7798	0.608
Right colon	7	1667.951	1410.0214		7	936.412	523.0055	
Left colon	28	2753.282	1996.2152	0.186	27	1758.973	1122.4103	0.071
Colon	24	2416.246	1841.0003		24	1519.279	1084.3813	
Rectum	11	2797.966	2169.4357	0.594	10	1758.445	1090.7798	0.563
**Ki-67 expression**								
negative	3	3617.660	2180.4306		3	1245.514	643.9262	
positive	32	2434.830	1907.2952	0.449	31	1622.923	1109.6303	0.441
**p53 expression**								
negative	12	3271.839	1679.5463		12	1526.599	861.8398	
positive	23	2152.412	1967.9908	0.09	22	1623.998	1193.4976	0.786

**Table 3 cancers-11-01536-t003:** Demographic, clinical, and pathological characteristics of the patients.

Characteristic	No.
**Patients**	
Cancer tissue	49
Adjacent normal tissues	49
**Sex**	
Male	28
Female	21
**Smoking**	
Yes	12
No	34
**BMI**	
<25	21
≥25	25
**Dukes staging system**	
B	25
C	19
D	5
**Histology grade**	
High	4
Medium	41
Low	4
**Tumor location**	
Right colon	8
Left colon	24
Rectum	17

BMI, Body Mass Index.

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
