# Peer review of "Understanding the Interplay between COX-2 and hTERT in Colorectal Cancer Using a Multi-Omics Analysis"

_cancers, 2019, doi:10.3390/cancers11101536_

Round 1
Reviewer 1 Report
In the current study, the authors studied COX-2, PGE2 levels and telomerase activity in CRC samples and concluded that COX-2 is inversely correlated with the telomerase activity. They also performed RNA-seq analysis with data extracted from the Cancer Genome Atlas. Overall, the manuscript is well written and reflects an exhaustive bioinformatics analysis, showing the experience of the authors with this type of approaching. However, I have some comments/suggestions:
Although the correlation value between COX-2 and telomerase activity is statistically significant, it is of low degree (R=-0.11). Authors should be more cautious when writing the title based on this result. This is a human tissue study. The approved protocol number for this study should be provided and this is absent in the text. In the figure 2A and 2C, the units are absent. They appear in the text but they should also appear in the graph. Also, the legend of figure 2A should be corrected. Considering that what is represented in the graph is not gene expression, PGE2 levels instead of PGE expression would be more correct. In the Table 2, it is not clear what the mean values represent. Please, try to put the units. In addition, the legend says “correlation” but the numbers do not seem to be correlation values.Author Response
We thank the reviewer for the insightful comments.
We have now revised our manuscript taking consideration all the reviewer comments:
1) We revised the title to “Interplay between COX-2 and hTERT in colorectal cancer”, and toned down the inverse correlation between COX-2 expression and telomerase activity in CRC, in the abstract, as suggested.
2) We have also added the units in the y-axes of Fig 2A & C, increased the font sizes in all axes, and revised the legend in Fig. 2A, accordingly (PGE2 levels instead of PGE expression).
3) In Table 2, the mean values represent PGE2 levels (pg/ml). Both Table 2 and its legend have now been revised, as per the reviewer’s request.
4) Further minor modifications have been made throughout the text after consulting an English native speaker.
We are hereby attaching the revised manuscript in track changes form.
Sincerely,
Apostolos Zaravinos, PhD
Associate Professor, Cancer Genetics
Department of Life Sciences
European University Cyprus
1516 Nicosia, Cyprus
[t] +357-22559577 [f] +357-22662051
[e] A.Zaravinos@euc.ac.cy [w] www.euc.ac.cy
Reviewer 2 Report
This manuscript by Ayiomamitis et al. focus on the implication of COX-2 in the early stages of colorectal cancer and in the interconnection between cyclooxygenase expression and telomerase function in this cancer. The paper is nicely written, the introduction and the results are composed in a competent style. The authors performed a rigorous analysis of the data obtained from the experiments and the conclusions are very interesting.
I have only a few misspellings and minor comments that should be corrected or considered before publication:
- Line 45: The term “worldwide” can be avoided or, for instance, you could rewrite the sentence: “Colorectal cancer is the third most common cancer worldwide among….”
- The quality of the figures is good, however figure legends 2 and 3 need to be structured (formatted)
- Materials and Methods: Please use the abbreviation “min” instead of minutes and always separate the number from the unit of measurement (100 ul instead of 100ul; 5ml should be 5 ml).
- Lines 573-607: Please delete all this
Author Response
We thank the reviewer for his judgment on our manuscript.
All minor comments have been now satisfied in the revised version of the manuscript.
Further minor modifications have also been made throughout the text after consulting an English native speaker.
Minor modifications are indicated in track-changes form of the attached revised version.
Sincerely,
Apostolos Zaravinos, PhD
Associate Professor, Cancer Genetics
Department of Life Sciences
European University Cyprus
1516 Nicosia, Cyprus
[t] +357-22559577 [f] +357-22662051
[e] A.Zaravinos@euc.ac.cy [w] www.euc.ac.cy